# Kynurenines and aerobic exercise capacity in chronic kidney disease: A cross-sectional and longitudinal study

Helena Wallin[1,2]*, Eva Jansson[1,2], Sophie Erhardt[3], Carin Wallquist[4], Britta Hylander[5], Stefan H. Jacobson[6], Kenneth Caidahl[2,7], Anette Rickenlund[2,7], Maria J. Eriksson[2,7]

1 Division of Clinical Physiology, Department of Laboratory Medicine, Karolinska Institutet, Stockholm, Sweden, 2 Department of Clinical Physiology, Karolinska University Hospital, Stockholm, Sweden, 3 Department of Physiology and Pharmacology, Karolinska Institutet, Stockholm, Sweden, 4 Department of Nephrology, Skåne University Hospital, Malmö, Sweden, 5 Department of Nephrology, Karolinska University Hospital, Karolinska Institutet, Stockholm, Sweden, 6 Division of Nephrology, Department of Clinical Sciences, Karolinska Institutet, Danderyd University Hospital, Stockholm, Sweden, 7 Department of Molecular Medicine and Surgery, Karolinska Institutet, Stockholm, Sweden

* helena.wallin@ki.se

**Data Availability Statement:** All relevant data are within the manuscript and its Supporting Information files.

## Abstract

### Background

The causes of reduced aerobic exercise capacity (ExCap) in chronic kidney disease (CKD) are multifactorial, possibly involving the accumulation of tryptophan (TRP) metabolites such as kynurenine (KYN) and kynurenic acid (KYNA), known as kynurenines. Their relationship to ExCap has yet to be studied in CKD. We hypothesised that aerobic ExCap would be negatively associated with plasma levels of TRP, KYN and KYNA in CKD.

### Methods

We included 102 patients with non-dialysis CKD stages 2–5 (CKD 2–3, n = 54; CKD 4–5, n = 48) and 54 healthy controls, age- and sex-matched with the CKD 2–3 group. ExCap was assessed as peak workload during a maximal cycle ergometer test. Plasma KYN, KYNA and TRP were determined by high-performance liquid chromatography. Kidney function was evaluated by glomerular filtration rate (GFR) and estimated GFR. The CKD 2–3 group and healthy controls repeated tests after five years. The association between TRP, KYN, KYNA and ExCap in CKD was assessed using a generalised linear model.

### Results

At baseline, there were significant differences between all groups in aerobic ExCap, KYN, KYNA, TRP and KYN/TRP. KYNA increased in CKD 2–3 during the follow-up period. In CKD 2–5, KYNA, KYN/TRP and KYNA/KYN were all significantly negatively associated with ExCap at baseline, whereas KYN and TRP were not. Kynurenines were significantly correlated with GFR (p < 0.001 for all). Including GFR in the statistical model, no kynurenines were independently associated with ExCap at baseline. At follow-up, the increase in KYN and KYN/TRP was related to a decrease in ExCap in CKD 2–3. After adjusting for GFR,

**Funding:** The study was supported by grants from the Stockholm County Council, grant numbers 2020 and 2021 received by MJE, and grants from the Swedish Kidney Foundation, grant number F2017–0048 received by AR, as well as grants from the Swedish Medical Research Council, grant number 2021-02251_VR received by SE. The funders had no role in study design, data collection and analysis, publication decisions, or manuscript preparation.

**Competing interests:** The authors have declared that no competing interests exist.

increase in KYN/TRP remained an independent significant predictor of a decline in ExCap in CKD 2–3.

## Conclusion

Aerobic ExCap was inversely associated with plasma levels of kynurenines in CKD at baseline and follow-up.

## Introduction

Aerobic exercise capacity is reduced in chronic kidney disease (CKD) and this is associated with increased morbidity and mortality, as well as a reduced quality of life [1–6]. The underlying mechanisms responsible for the decrease in exercise capacity in individuals with CKD are multifaceted and related to the complex pathogenesis of CKD. Both central and peripheral factors might be important determinants of aerobic exercise capacity in CKD [7, 8]. With the progressive loss of kidney function, metabolic waste products and uraemic substances accumulate, leading to increased oxidative stress, chronic inflammation and mitochondrial dysfunction [9–11]. Such mechanisms may explain the deterioration of organ system function in CKD and, ultimately, the reduced exercise capacity.

One substance that accumulates is kynurenine (KYN), a metabolite of the essential amino acid tryptophan (TRP) [9, 12]. The majority of free TRP is metabolised to KYN along the KYN pathway. KYN is transformed to either kynurenic acid (KYNA) through kynurenine aminotransferase (KAT) or to other metabolites, mainly quinolinic acid [9, 12–14]. TRP and metabolites of the KYN pathway, known as kynurenines, are modulators of biological processes such as systemic inflammation and cardiovascular function, and have been linked to atherosclerosis and muscle damage in CKD [9, 15].

Kynurenines have been associated with frailty in the elderly [16, 17]. In addition, plasma KYN levels in heart failure have been negatively correlated with peak oxygen uptake and muscle strength and positively associated with mortality [18, 19]. However, the relationship between kynurenines and exercise capacity in CKD has not been reported. Given the accumulation of kynurenines in CKD, we investigated the association between aerobic exercise capacity and kynurenines in non-dialysis individuals with CKD. We hypothesised that aerobic exercise capacity would be negatively associated with plasma levels of KYN, KYN/TRP and KYNA in CKD. This was tested in the current study using cross-sectional and longitudinal analyses.

## Materials and methods

### Study population and protocol

The current study population is part of the PROGRESS 2002 study, a single-center, prospective observational cohort study conducted at Karolinska University Hospital in Stockholm, Sweden. Cross-sectional and longitudinal analyses were performed [4, 20–22]. Subjects who provided blood samples for TRP, KYN and KYNA analyses and who were scheduled for exercise tests were included. Blood sampling and exercise tests were conducted at the most a few days apart. At baseline, there were 102 patients with stages 2–5 of non-dialysis CKD (stages 2–3, n = 54 patients; stages 4–5, n = 48 patients), and 54 age- and sex-matched healthy controls (Fig 1). The age at inclusion was 18–65 years. The patients were consecutively recruited from the Department of Nephrology at Karolinska University Hospital from September 18th 2002–June 11th

| | CKD 2-3 GFR 50-70 mLmin/1.73 m² | CKD 4-5 GFR <20 mL/min/1.73 m² | Controls GFR >80 mL/min/1.73 m² | |
|---|---|---|---|---|
| **Baseline** | n = 54 | n = 48 | n = 54 | Excluded outliers kynurenines: CKD 2-3: 1 KYNA, 2 TRP samples CKD 4-5: 3 TRP samples Controls: 1 KYN, 1 TRP sample |
| Blood samples kynurenines | n = 53 | n = 48 | n = 53 | |
| Exercise test | n = 52 | n = 47 | n = 54 | |
| Kynurenines and exercise test | n = 51 | n = 46 | n = 53 | |
| **5-year follow-up** | n = 49 | No follow-up | n = 43 | Lost to follow-up CKD 2-3: 5 individuals Controls: 11 individuals |
| Blood samples kynurenines | n = 48 | | n = 42 | |
| Exercise test | n = 45 | | n = 40 | Excluded outliers kynurenines: CKD 2-3: 2 KYNA samples, |
| Kynurenines and exercise test | n = 44 | | n = 36 | |

**Fig 1. Study population and numbers of performed exercise tests and kynurenine analyses at baseline and the 5-year follow-up.** Numbers of individuals lost to follow-up and excluded outliers are shown. KYN = kynurenine, KYNA = kynurenic acid, TRP = tryptophan.

2009 during outpatient visits. The control group was recruited either by random selection from the Swedish Total Population Register or by advertisement at the Karolinska University Hospital website. Baseline data was collected for the CKD 2–3 group during 2002–2003, for CKD 4–5 during 2002–2009, and for the control group during 2004–2007. The inclusion criteria were based on the glomerular filtration rate (GFR) with GFR 50–70 mL/min for CKD 2–3, GFR <20 mL/min for CKD 4–5 and GFR >80 mL/min for controls [23]. The exclusion criteria for the controls were a history of kidney disease, diabetes or cardiovascular disease, and any ongoing medication. The exclusion criteria for all subjects were active infection, current immunosuppressive therapy with steroids or cytotoxic drugs, current malignancy, kidney transplantation or kidney donation, and blood-transmitted disease.

All three groups were examined at inclusion, that is, baseline (cross-sectional comparison). Assessments were repeated after five years in the CKD 2–3 group and the healthy control group. Additionally, the CKD 2–3 group was regularly followed in a nephrology clinic to aggressively treat hypertension, hyperlipidaemia and proteinuria. The final population of the current study at baseline and 5-year follow-up, including subjects lost to follow-up, is shown in Fig 1.

The study protocol was reviewed and approved by the Local Ethics Committee and Institutional Review Board of the Karolinska Institutet at the Karolinska University Hospital. All participants gave their informed consent in writing.

## Blood samples

Venous blood samples were obtained in fasting state at baseline and at the 5-year follow-up visit. Plasma was prepared and stored at –70°C until analysis. The concentrations of haemoglobin, creatinine, and high-sensitivity C-reactive protein (hs-CRP) were analysed at the Karolinska University Laboratory according to routine protocols.

### Kidney function

GFR was determined by plasma clearance of iohexol in all subjects at baseline [24]. Iohexol-measured GFR was repeated in the CKD 2–3 group at the 5-year follow-up but not in controls. Therefore, for longitudinal comparison between CKD stages 2–3 and controls, the Chronic Kidney Disease-Epidemiology Collaboration (CKD-EPI) equation was used to calculate the estimated GFR (eGFR) through creatinine [25].

### Plasma measurement of TRP, KYN and KYNA

TRP, KYN and KYNA measurements were performed by high-performance liquid chromatography (HPLC) with ultraviolet and fluorescence detection, as previously described [26]. Plasma aliquots were treated with perchloric acid (0.4 M, containing 0.1% sodium metabisulfite and 0.05% EDTA) and centrifuged. The centrifugation procedure was repeated, followed by an additional 70% perchloric acid (10% by volume). Then, 20 μL of supernatant was injected into an HPLC column (ReproSil 100 C18, 3-μm particles, 100 × 4 mm, Dr. Maisch, GmbH, Germany) with acetonitrile/sodium acetate (6.8%/30 mM, pH 6.2 adjusted with acetic acid) as eluent at a flow rate of 0.5 mL/min. KYN detection was achieved with an ultraviolet detector set at a wavelength of 360 nm and TRP detection at a wavelength of 240 nm. The eluate was further mixed online with zinc acetate (final concentration 0.5 M). KYNA was determined with a fluorescence detector set at an excitation wavelength of 344 nm and an emission wavelength of 398 nm. Signals from the detectors were transferred to a computer for analysis with Datalys Azur (version 4.6.0.0). The TRP, KYN and KYNA concentrations were extrapolated from standard curves prepared daily from mixtures of reference solutions. The detection limit for each assay was at least 20 times lower than the plasma concentrations reported. The retention times for TRP, KYN and KYNA were approximately 7.3, 4.2 and 7.0 min, respectively. The reliability of the method was verified by running some samples in duplicate; the coefficients of variation were 5–7% for TRP, KYN and KYNA.

### Aerobic exercise capacity, physical activity and echocardiography

Exercise testing was performed as previously described [4]. In brief, it comprised an incremental test on a cycle ergometer (RE990; Rodby Innovation AB, Uppsala, Sweden) with a protocol consistent with current clinical standards. All tests were driven to volitional exhaustion, and participants rated the highest perceived exertion according to the Borg CR10 scale. The peak workload in watts (W) was used to measure aerobic exercise capacity. The results of exercise tests at baseline and at the 5-year follow-up have been reported [4, 20]. Predicted values for peak workload were based on a Swedish population study that considers age, sex, height and workload increment per minute [27].

The physical activity level was rated on a four-point scale modified from the Saltin–Grimby Physical Activity Level Scale [28]. Echocardiography was performed in all patients (Sequoia 512; Siemens Medical Solutions, Mountain View, CA, USA) according to current guidelines for assessing cardiac function [29]. Echocardiography assessed the left ventricular (LV) ejection fraction as a measure of systolic LV function and the ratio of early mitral filling velocity and early diastolic myocardial velocity (E/e′ ratio) as a measure of diastolic LV function.

### Statistical analysis

Descriptive data are presented as numbers, percentages, mean and standard deviation, or median and interquartile ranges for skewed variables. We assessed the validity of extreme outliers, defined as a data point that lies more than three times the interquartile range above the

third quartile or below the first quartile. Welch's test, one-way ANOVA with post hoc tests, the chi-square test and Kruskal–Wallis test, where applicable, were performed to compare the groups at baseline. Generalised linear models (GLM) were mainly used to analyse the association between variables. When peak workload was used as the dependent variable, adjustment for sex, age and height squared was performed as known non-disease-related determinants of aerobic exercise capacity [27, 30]. Where applicable, the Spearman correlation coefficient ($r_s$) was used to analyse the relationship between two variables. A p-value < 0.05 for a two-tailed test was defined as statistically significant. Statistical analyses were performed using IBM SPSS Statistics (version 28.0; IBM, Armonk, NY, USA).

## Results

### Subject characteristics

The baseline characteristics of the study population are presented in Table 1. The aetiology of CKD included familial/hereditary/congenital disease (n = 30), primary glomerulonephritis

**Table 1. Subject characteristics for the whole study population.**

| Variable | Controls | CKD stages 2–3 | CKD stages 4–5 | p-value *(overall)* | p-value *(post hoc)** |
|---|---|---|---|---|---|
| Subjects, (n) | 54 | 54 | 48 | | |
| Age, (years) | 48 ± 11 | 47 ± 11 | 49 ± 12 | 0.6 | |
| Male, n (%) | 33 (61) | 33 (61) | 29 (60) | 1.0 | |
| Height, (cm) | 176 ± 9 | 174 ± 9 | 174 ± 10 | 0.5 | |
| Weight, (kg) | 77 ± 12 | 77 ± 19 | 77 ± 17 | 1.0 | |
| BMI (kg/m$^2$) | 25 ± 3.4 | 25 ± 4.8 | 25 ± 4.0 | 0.7 | |
| Lean body mass, (kg) | 54 ± 1.1 | 52 ± 1.1 | 50 ± 1.3 | 0.3 | |
| GFR, (mL/min/1.73 m$^2$) | 99 ± 13 | 60 ± 5.2 | 15 ± 3.8 | < 0.001 | < 0.001 for all |
| eGFR, (mL/min/1.73 m$^2$) | 96 ± 13 | 58 ± 13 | | < 0.001 | |
| Haemoglobin, (g/dL) | 14.2 ± 1.2 | 13.6 ± 1.4 | 12.2 ± 1.3 | < 0.001 | 0.008/ 0.001/0.001 |
| hs-CRP, (mg/L) | 0.89 (0.46–2.1) | 1.8 (0.98–4.0) | 1.6 (0.94–3.2) | 0.007 | 0.007/0.08/1.0 |
| Diabetes, n (%) | | 11 (21) | 7 (15) | 0.4 | |
| *Physical activity self-reported* | | | | | |
| Physical activity level** | 2 (1–3) | 3 (2–3) | 3 (2.25–3) | < 0.001 | 0.08/ < 0.001/0.3 |
| *Cardiac function* | | | | | |
| LV ejection fraction, (%) | 65 ± 8.4 | 62 ± 7.6 | 62 ± 9.7 | 1.0 | |
| LV diastolic function, (E/e′) | 5.0 ± 1.2 | 5.6 ± 1.4 | 6.3 ± 1.6 | < 0.001 | 0.1/ < 0.001/0.05 |
| *Medication* | | | | | |
| Beta-blocker, n (%) | | 11 (20) | 18 (38) | 0.06[a] | |
| Diuretics, n (%) | | 12 (22) | 33 (59) | < 0.001[a] | |
| ACE inhibitors, n (%) | | 23 (43) | 28 (53) | 0.1[a] | |
| Angiotensin II blocker, n (%) | | 22 (41) | 23 (48) | 0.5[a] | |
| Calcium-channel blocker, n (%) | | 10 (19) | 29 (60) | < 0.001[a] | |
| Lipid-lowering therapy, n (%) | | 13 (24) | 31 (65) | < 0.001[a] | |

Values are reported as number (percentage) or mean ± standard deviation. Median (interquartile range) reported for skewed variables. Kruskal–Wallis test with

Bonferroni correction for comparison between groups for skewed variables. P-values for between-group comparisons with Tukey post hoc test if not otherwise stated.

n = number of subjects and indicates the maximal number of subjects per group. Because of some missing values, the total number for each variable varies.

* CKD 2–3 vs. controls/CKD 4–5 vs. controls/CKD 2–3 vs. CKD 4–5.

[a]Chi-square test

** Level of physical activity; 1 most active. BMI = body mass index, LV = left ventricle, E/e′ = ratio of early mitral filling velocity/early diastolic myocardial velocity.

(n = 26), secondary glomerular/systemic disease (n = 18), and miscellaneous/unknown cause (n = 28). There were no significant differences in age, sex or body size between any of the groups, healthy controls, individuals with CKD 2–3 or CKD 4–5. There were significant differences between all groups in GFR, as expected, and in haemoglobin levels, with the lowest GFR and haemoglobin levels in CKD 4–5. The level of physical activity was significantly lower in CKD 4–5 than in controls, and hs-CRP was significantly higher in CKD 2–3 compared with controls. Cardiac systolic function was normal in all groups. Cardiac diastolic function, as estimated by E/e′, differed significantly between the CKD 4–5 group and healthy controls. However, the differences were of subclinical magnitude (Table 1).

## Cross-sectional comparison at baseline

Fig 2 presents the results of aerobic exercise capacity expressed as peak workload and kynurenines in individuals with CKD 2–3, CKD 4–5 and controls at baseline. Aerobic exercise capacity differed significantly between all three groups. The lowest exercise capacity was found in CKD 4–5 and the highest in controls. There were also significant differences between all

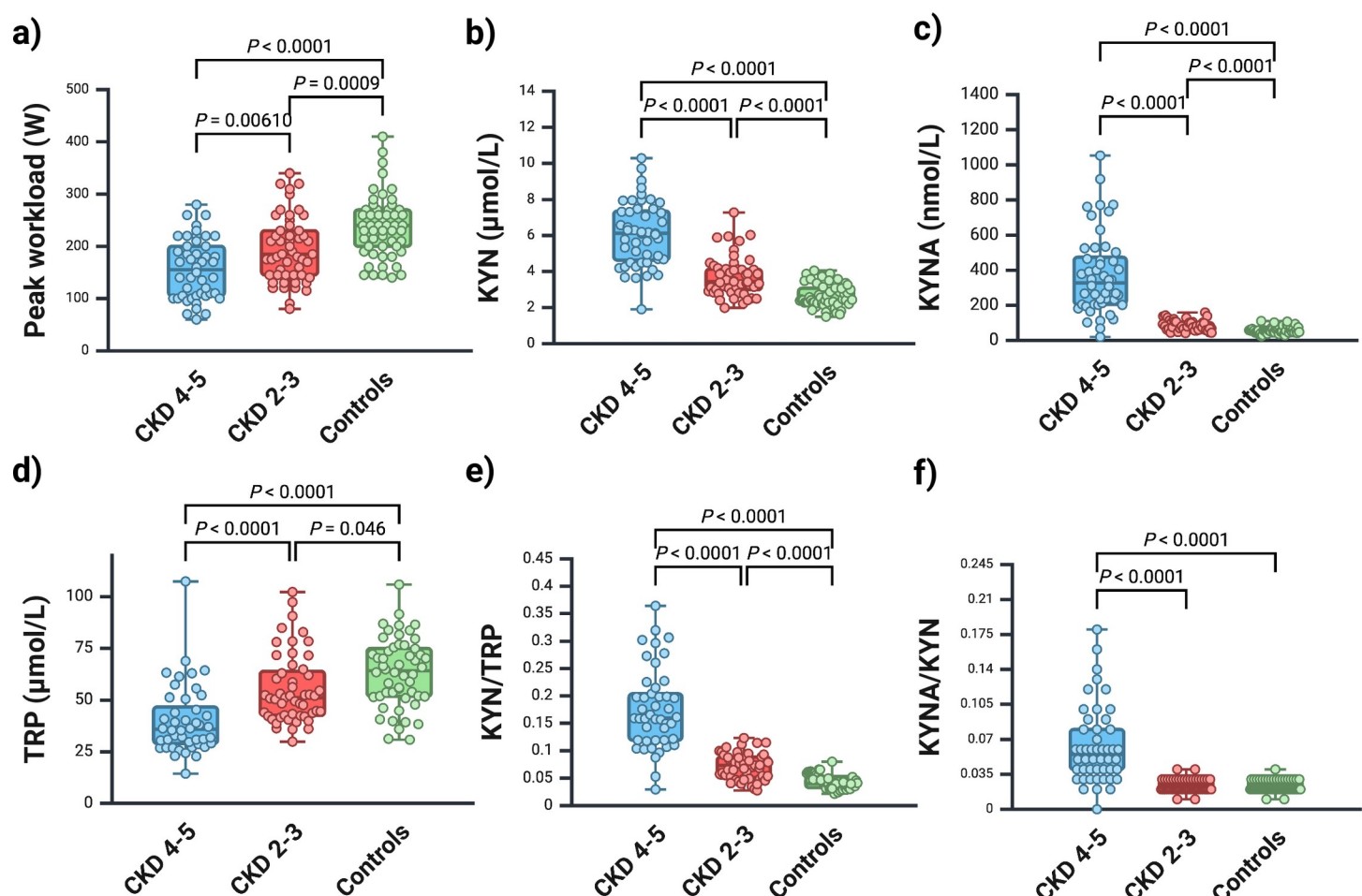

**Fig 2.** a-f. Box plots of peak workload, TRP, and kynurenines in CKD 2–3, CKD 4–5 and controls at baseline. Whiskers represent highest through lowest values, boxes the interquartile range and circles individual values. a: p-value by one-way ANOVA with Tukey post hoc test, b-f: p-values by one-way ANOVA with Welch's test and Dunnett's T3 post hoc test. KYN = kynurenine, KYNA = kynurenine acid, TRP = tryptophan. Number of subjects: 45–53. Created in BioRender. Wallin, H. (2024) https://BioRender.com/h85q062.

groups in TRP, KYN, KYNA and KYN/TRP. The KYNA/KYN ratio differed significantly between CKD 2–3 and CKD 4–5, and between CKD 4–5 and controls, but not between CKD 2–3 and CKD 4–5 (Fig 2).

The association between aerobic exercise capacity and kynurenines at baseline in individuals with stages 2–5 of CKD are shown in Fig 3A. We used a GLM for this analysis, which included known exercise capacity determinants (age, sex and height). KYN, KYNA, TRP, KYN/TRP and KYNA/KYN were tested one by one in a separate model adjusting for age, sex and height. KYNA, KYN/TRP and KYNA/KYN were significantly inversely associated with exercise capacity, while KYN and TRP were not (Fig 3A). When GFR was included in each model, none of the kynurenines were independently associated with exercise capacity, but KYN was borderline significant (p = 0.06). Details from the GLMs can be found in S1 Table. GFR was significantly (p < 0.001 for all comparisons) correlated with all kynurenines (KYN ($r_s$) = –0.6, KYNA ($r_s$) = –0.7, TRP ($r_s$) = 0.4, KYN/TRP ($r_s$) = –0.7, and KYNA/KYN ($r_s$) = –0.6), suggesting a strong covariation. The relationship between individual KYN/TRP values and achieved workload as a percentage of predicted peak workload in all three groups is shown in Fig 4. TRP did not correlate with age in any of the groups ($r_s$ = -0.07 and p = 0.6 for CKD 2–3, $r_s$ = -0.3 and p = 0.07 for CKD 4–5, $r_s$ = -0.2 and p = 0.2 for the controls).

### Longitudinal comparison: A 5-year follow-up

The CKD 2–3 group and healthy controls were followed and re-examined after five years. Table 2 shows the baseline and 5-year follow-up data for TRP and kynurenines of both groups. Aerobic exercise capacity did not change significantly from baseline to 5-year follow-up, neither for CKD 2–3 (p = 0.06) nor for controls (p = 0.3).

KYNA increased significantly during the follow-up period in CKD 2–3, and unexpectedly there was an increase in TRP. There was a trend towards higher levels of KYNA/KYN in CKD 2–3 (p = 0.08). The KYN/TRP ratio decreased in controls, driven by the increase in TRP.

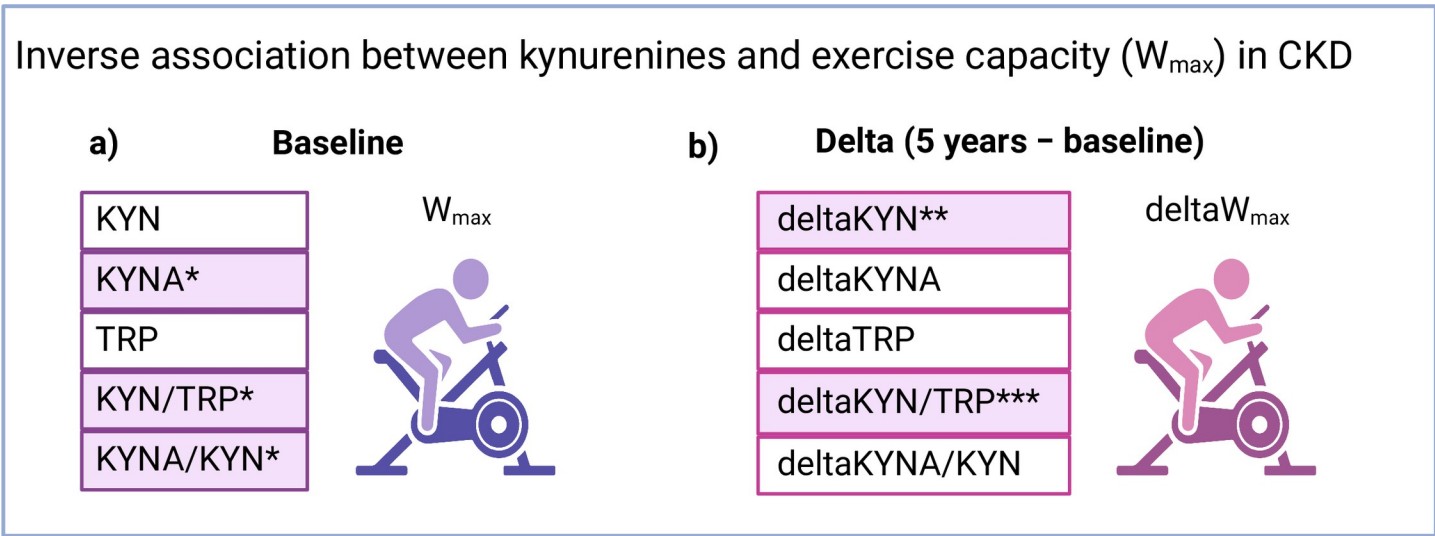

**Fig 3.** a) Association between kynurenines, TRP and aerobic exercise capacity (expressed as peak workload) in CKD 2–5. b) Association between the change in kynurenines, TRP and aerobic exercise capacity over 5 years in CKD 2–3. *p < 0.05, **p < 0.01, ***, p < 0.001 denotes significant inverse association. Assessed by GLM (generalised linear model) with age, sex and height included in all baseline models., KYN = kynurenine, KYNA = kynurenic acid, TRP = tryptophan, $W_{max}$ = peak workload on maximal exercise test. Created in BioRender. Wallin, H. (2025) https://BioRender.com/i04z472.

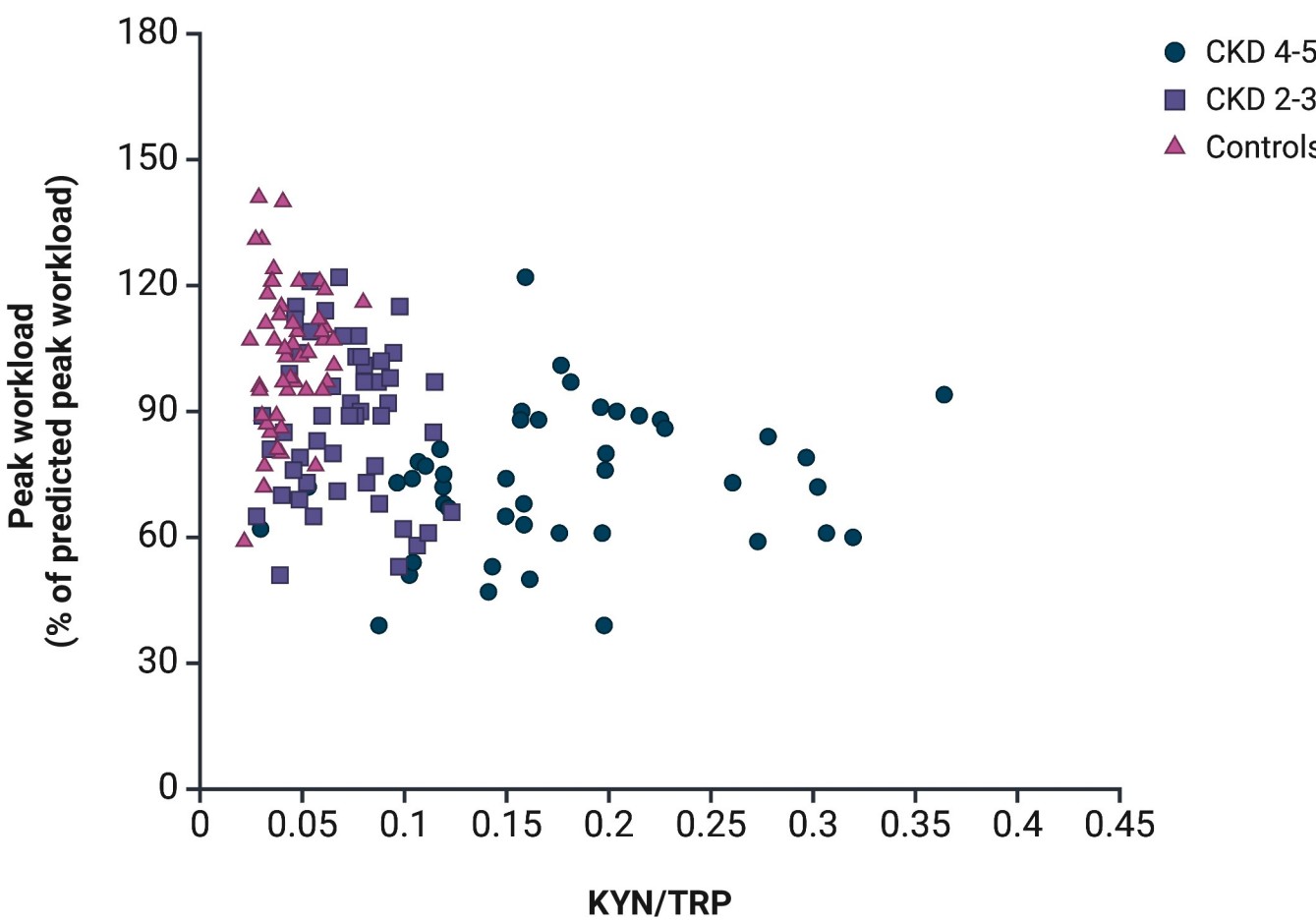

**Fig 4. The relationship between aerobic exercise capacity expressed as peak workload as a percentage of predicted peak workload and KYN/TRP ratio, divided by CKD 2–3, CKD 4–5 and controls.** KYN = kynurenine, TRP = tryptophan. Created in BioRender. Wallin, H. (2025) https://BioRender.com/y91k603.

**Table 2. Aerobic exercise capacity, kynurenines and TRP at baseline and at 5 years follow-up, and a comparison between CKD 2–3 and controls regarding the change in these variables.**

| Variable | Controls | | | CKD stages 2–3 | | | Delta values | | p-value[δ] |
|---|---|---|---|---|---|---|---|---|---|
| | Baseline | 5-year follow-up | p-value[*] | Baseline | 5-year follow-up | p-value[*] | Controls | CKD | |
| Peak workload (W) | 246 ± 63 | 241 ± 70 | 0.3 | 199 ± 60 | 193 ± 61 | 0.06 | -4.6 ± 29 | -5.7 ±20 | 0.8 |
| KYN (µmol/L) | 2.7 ± 0.7 | 2.6 ± 0.5 | 0.7 | 3.7 ± 1.2 | 4.0 ± 1.3 | 0.2 | -0.04 ± 0.74 | 0.32 ± 1.6 | 0.2 |
| KYNA (nmol/L) | 60 ± 19 | 59 ± 16 | 0.7 | 85 ± 26 | 100 ± 40 | 0.01 | -1± ±17 | 14 ± 38 | 0.01 |
| TRP (µmol/L) | 64 ± 17 | 68 ± 11 | 0.08 | 57 ± 17 | 63 ± 12 | 0.045 | 4.2 ± 15 | 5.6 ± 18 | 0.7 |
| KYN/TRP | 0.044 ± 0.1 | 0.039 ± 0.008 | 0.01 | 0.068 ± 0.03 | 0.066 ± 0.03 | 0.7 | -0.004 ± 0.01 | -0.002 ± 0.03 | 0.6 |
| KYNA/KYN | 0.022 ± 0.006 | 0.022 ± 0.005 | 1.0 | 0.024 ± 0.006 | 0.026 ± 0.009 | 0.08 | -0.04 ± 6.2 | 1.9 ± 8.4 | 0.13 |

Values reported as mean ± standard deviation. Delta value = 5-year follow-up–baseline.

[*]Paired t-test for differences between baseline and 5-year follow-up for each group.

[δ]Comparison of changes from baseline to the 5-year follow-up between CKD 2–3 and controls assessed by GLM (generalised linear model). KYN = kynurenine, KYNA = kynurenic acid, TRP = tryptophan. Number of subjects = 39–50.

Changes in kynurenines over five years differed significantly between patients with CKD and controls only for KYNA (Table 2).

eGFR expressed in mL/min/1.73 m$^2$ decreased significantly, both in patients with CKD 2–3 and in controls, but remained within the normal range in controls (58 ± 13 at baseline vs 50 ± 17 at follow-up for CKD 2–3 and 94 ± 12 at baseline vs 91 ± 11 at follow-up for controls). Haemoglobin increased slightly but significantly in both groups (13.5 ± 1.4 g/dL at baseline vs 13.9 ± 1.3 g/dL at follow-up for CKD 2–3 and 14.2 ± 1.1 g/dL at baseline vs 14.7 ± 1.1 g/dL at follow-up for controls). No correlation was found between the increase in TRP and haemoglobin levels in CKD 2–3 (p = 0.3). Albumin did not increase significantly in CKD 2–3 (37.7 g/L at baseline vs. 37.9 g/L at follow-up) or in the control group (39.5 g/L at baseline vs. 40.0 g/L at follow-up) (p = 0.2 for both groups). Subjects lost to follow-up, in both the CKD 2–3 and control group, did not differ significantly (p > 0.05) regarding age, GFR and ExCap, from those attending follow-up.

The association between 5-year changes in aerobic exercise capacity and kynurenines in patients with CKD 2–3 is presented in Fig 3B. Changes (follow-up vs. baseline) in KYN and KYN/TRP were significantly and inversely associated with the corresponding change in exercise capacity in CKD 2–3). However, there were no significant associations between changes in KYNA (p = 0.06), KYNA/KYN and TRP and change in exercise capacity. With the inclusion of GFR, only the change in KYN/TRP remained a significant predictor of the change in exercise capacity (p = 0.02, n = 40), whereas the change in KYN was borderline significant (p = 0.05, n = 41). Details from these analyses can be found in S2 Table.

## Discussion

This is the first study to report the relationship between aerobic exercise capacity and kynurenines in CKD.

### Main findings

Higher levels of KYNA and the KYN/TRP and KYNA/KYN ratios in non-dialysis stages 2–5 of CKD were all associated with lower aerobic exercise capacity. Furthermore, in individuals with CKD 2–3, an increase in KYN and KYN/TRP was associated with a decrease in exercise capacity over five years. The hypothesis that aerobic exercise capacity would be inversely associated with plasma levels of kynurenines in CKD was thus confirmed in both the cross-sectional and longitudinal design.

### Kynurenines and exercise capacity

KYN and the KYN/TRP ratio have been associated with frailty [16, 17, 31] and muscle atrophy [32]. KYN has also been suggested as a possible biomarker for reduced muscle endurance in heart failure patients [33]. Furthermore, a study on heart failure reported a negative correlation between KYN and functional capacity measured as peak oxygen uptake and with the 6-minute walk test [18]. However, the relationship between KYN and peak oxygen uptake did not persist when adjusted for creatinine, suggesting that impaired kidney function and a related increase in KYN influenced exercise capacity in heart failure. Similarly, in our study, the association between kynurenines and exercise capacity at baseline was not significant when adjusted for GFR. One reason for this may be a strong covariation between kynurenines and GFR. The reduction in exercise capacity may be influenced by the biological effects of increased levels of KYN and KYNA, caused by the declining renal function. In our study, kynurenines were correlated with GFR, and GFR was a strong predictor of exercise capacity. After five years however, the KYN/TRP ratio remained a significant predictor of change in

exercise capacity in CKD even when adjusted for GFR, suggesting that there is an independent relationship between kynurenines and exercise capacity in CKD. The distribution of GFR at baseline in the CKD 2–5 group differed from the kynurenines as the two merged CKD groups were based on separate GFR ranges. Therefore, it was not surprising that GFR had such a strong and independent association with exercise capacity at baseline, while in the analyses of the change in exercise capacity, KYN/TRP was possibly even more strongly associated with exercise capacity than GFR.

## Kynurenines and kidney function

Our results on baseline levels of kynurenines agree with other studies in CKD that report low levels of TRP and high levels of KYN, KYNA and KYN/TRP [10, 13, 34, 35]. The increased activity of two key enzymes that convert TRP to KYN, indoleamine 2,3-dioxygenase and tryptophan 2,3-dioxygenase, is linked to increased KYN levels and decreased TRP levels in CKD [35, 36]. Changes of TRP and KYN in blood have been suggested as early biomarkers for CKD. The ratio of KYN to TRP may be an even more sensitive marker for kidney function [10]. It has been shown to have a prognostic value in the general population [37], and recently also in diabetes [38] and in kidney disease [39]. In addition to an increase in enzyme activity caused mainly by chronic inflammation, a reduction in GFR and decreased tubular secretion contribute to the increase in serum levels of kynurenines [9, 12].

Although there were significant differences between the CKD group and healthy controls in plasma levels of TRP, KYN and KYNA at baseline, only KYNA increased significantly during the follow-up period in CKD 2–3. This implies that KYNA may be a more sensitive marker than KYN for subtle disease progression in CKD, considering that our 5-year decrease in GFR was modest. This finding is consistent with a previous longitudinal study in a wide range of patients with CKD, in which KYNA levels showed the strongest association with the progression of CKD among several studied metabolites [40]. However, there are also conflicting results, given that another study on moderate CKD failed to show any relationship between KYNA or KYN and disease progression [41]. KYNA, which is dependent on tubular secretion and glomerular filtration for clearance, is one of the end-products of the KYN pathway [42], which may explain its increased accumulation compared with KYN as kidney damage progresses. Dysregulation of the KYN pathway is seen in several chronic diseases [43, 44], with the typical feature being an increase in KYN, KYN/TRP and a decrease in KYNA [45–49]. However, in CKD, dysregulation of the KYN pathway differs from that seen in other chronic conditions, especially because of the increase in KYNA. It is worth noting that patients in the CKD 2–3 group in our study underwent intensive treatment for cardiovascular risk factors and were closely followed at a nephrology clinic, which could explain the modest changes in kynurenines over time in this group.

An unexpected finding in our study was that plasma TRP concentrations increased slightly over five years in the CKD 2–3 and control groups. Blood samples were obtained in fasting state at both visits and analysed during the same session, which should reduce method-dependent variation. Low TRP levels have been previously associated with low haemoglobin levels and iron deficiency [50]. In the current study, there was no significant correlation between the increase in TRP and the increase in haemoglobin level, although the changes in both variables were quite small. As albumin did not increase significantly during the follow-up period, haemoconcentration is unlikely to explain the results. Likewise, age seems an implausible explanation for the increase in TRP over five years, as there were no correlations between TRP and age for patients with CKD 2–3 or the control group at baseline.

## Potential link between kynurenines and aerobic exercise capacity

Both central and peripheral factors may influence aerobic exercise capacity in CKD. Reduced cardiac function and systemic oxygen delivery, anaemia, vascular stiffness, abnormal neurocirculatory control, myopathy and reduced muscle strength are all associated with reduced exercise capacity in CKD [4, 5, 8, 51–56]. Although the mechanistic link between kynurenines and exercise capacity is unknown, mitochondrial dysfunction could be a mediator. Kynurenines can induce mitochondrial dysfunction through increased oxidative stress [57]. From earlier *in vitro* studies, negative effects of kynurenines on mitochondria have been demonstrated in both heart and skeletal muscle [11, 58–63]. KYN, KYNA and other uraemic metabolites were found to alter mitochondrial energetics of skeletal muscle in mice [62]. Mitochondrial dysfunction is believed to contribute to both uraemic myopathy [64–66] and cardiomyopathy [67]. Thus, mitochondrial dysfunction can affect both skeletal and cardiac muscle tissue, resulting in impaired function and decreased aerobic exercise capacity. Several studies have also reported a link between kynurenines and cardiovascular disease, in particular atherosclerosis [68–70], which is frequently present in CKD.

## Clinical implications

Reduced exercise capacity is associated with increased mortality and morbidity in CKD and is often accompanied by muscle weakness and fatigue. A better understanding of the underlying mechanisms may help develop targeted interventions, whether medical therapy or exercise. The metabolites of the KYN pathway may be an important link to cardiovascular and muscular dysfunction in CKD, resulting in exercise intolerance and poor quality of life. This study did not test the clinical relevance of the results and larger studies are needed to assess that. Even though the cohort did not have severe cardiovascular disease, some of the patients had diabetes. While it would be interesting to explore the influence of comorbidities on the results, this would also require a larger study.

## Strengths, limitations and future perspectives

No previous study has reported the relationship between aerobic exercise capacity and kynurenines in CKD. The strengths of this study are the study population, including both men and women and a matched control group, and the prospective study design with a 5-year follow-up, allowing cross-sectional and longitudinal comparisons. Our findings could be applicable to the broader population of non-dialysis CKD as it includes a large span of GFR, ranging from mild to severe disease, and the cohort also includes different etiologies of CKD. However, the follow-up data is limited to the CKD 2–3 group, and therefore the results might be more applicable to this population.

One limitation is the lack of standardisation of diet or physical exercise in the days before blood sampling, which may have influenced TRP, KYN and KYNA levels [71]. However, the blood samples were taken in a fasting state. Another limitation is that further downstream KYN metabolites such as 3-hydroxy-kynurenine, picolinic acid and quinolinic acid were not analysed. We did not analyse the influence of medications on kynurenines; however, those analyses would be quite challenging because of the covariation between disease progression and medication use. No specific sample size calculation was performed, however the current study of the association between ExCap and kynurenines was of an exploratory character. The controls were generally recruited at a later stage than the CKD group, but the examinations were carried out according to the same protocols during the whole study period. Recruitment bias is always a possibility in studies assessing exercise capacity, as e.g. sedentary patients are less likely to choose to take part. A further limitation is that a cause-and-effect relationship

between exercise capacity and kynurenines cannot be confirmed, as the mechanistic links were not explored in the current study. A further step to explore the mechanistic links would be to combine kynurenines and measurement of exercise capacity with muscle biopsies to study mitochondrial function. In non-CKD populations, kynurenine levels are influenced by acute and chronic exercise [72–74]. Therefore, future larger studies that clarify the effect of exercise on kynurenines in CKD could contribute to an increased understanding of the impact of kynurenines on exercise capacity.

## Conclusion

Aerobic exercise capacity was negatively associated with plasma kynurenine levels in non-dialysis CKD, both cross-sectionally at baseline and in the 5-year follow-up study. However, more research is needed to establish the role of kynurenines and to identify mechanisms, for example, mitochondrial dysfunction, that may link kynurenines with reduced aerobic exercise capacity in CKD.

## Supporting information

**S1 Table. Association between aerobic exercise capacity and TRP, kynurenines, TRP and GFR in CKD 2–5 at baseline.**
(DOCX)

**S2 Table. Predictors of the change in aerobic exercise capacity from baseline to 5-year follow-up in CKD 2–3.**
(DOCX)

**S1 File. Group comparisons with outliers.**
(PDF)

**S2 File. Data file without outliers.**
(XLSX)

## Acknowledgments

We thank the staff and participants of the PROGRESS study, especially Agneta Aspegren-Pagels, PhL, MSc, for her important contributions and Anna Asp, MD, PhD, for the collaboration on cardiovascular data. The authors acknowledge Professor Matteo Bottai for assistance with the statistical analysis of the data and Dr Michel Goiny for assistance with HPLC.

## Author Contributions

**Conceptualization:** Helena Wallin, Eva Jansson.

**Formal analysis:** Helena Wallin, Eva Jansson, Anette Rickenlund.

**Investigation:** Helena Wallin, Carin Wallquist, Britta Hylander, Stefan H. Jacobson, Kenneth Caidahl, Maria J. Eriksson.

**Methodology:** Helena Wallin.

**Project administration:** Carin Wallquist, Britta Hylander, Stefan H. Jacobson, Kenneth Caidahl, Maria J. Eriksson.

**Resources:** Sophie Erhardt.

**Supervision:** Eva Jansson, Sophie Erhardt, Anette Rickenlund, Maria J. Eriksson.

**Writing – original draft:** Helena Wallin.

**Writing – review & editing:** Helena Wallin, Eva Jansson, Sophie Erhardt, Stefan H. Jacobson, Kenneth Caidahl, Anette Rickenlund, Maria J. Eriksson.

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
