## [Decision Letter · Decision Letter 0]

10 Jul 2024

PONE-D-24-08103Kynurenines and aerobic exercise capacity in chronic kidney disease: a cross-sectional and longitudinal studyPLOS ONE

Dear Dr. Wallin,

Thank you for submitting your manuscript to PLOS ONE. After careful consideration, we feel that it has merit but does not fully meet PLOS ONE’s publication criteria as it currently stands. Therefore, we invite you to submit a revised version of the manuscript that addresses the points raised during the review process.

We look forward to receiving your revised manuscript.

Kind regards,

Francesco Fazio

Academic Editor

PLOS ONE

Journal Requirements:

Reviewers' comments:

Reviewer's Responses to Questions

**Comments to the Author**

1. Is the manuscript technically sound, and do the data support the conclusions?

Reviewer #1: Yes

Reviewer #2: Yes

Reviewer #3: Yes

2. Has the statistical analysis been performed appropriately and rigorously? 

Reviewer #1: Yes

Reviewer #2: Yes

Reviewer #3: Yes

3. Have the authors made all data underlying the findings in their manuscript fully available?

Reviewer #1: Yes

Reviewer #2: Yes

Reviewer #3: Yes

4. Is the manuscript presented in an intelligible fashion and written in standard English?

Reviewer #1: Yes

Reviewer #2: Yes

Reviewer #3: Yes

5. Review Comments to the Author

Reviewer #1: Thank you for the opportunity to review this manuscript. Overall, this study provides a thorough examination of the relationship between aerobic exercise capacity and kynurenines in non-dialysis chronic kidney disease (CKD) patients. I appreciate the authors' dedication to undertaking this significant study.

1. Is the manuscript technically sound, and do the data support the conclusions?

Yes. Overall, the manuscript appears technically sound, and the data presented support the conclusions drawn by the authors. The study design provides robust evidence for the association between plasma kynurenine levels and aerobic exercise capacity in non-dialysis CKD patients. The statistical analyses performed, such as general linear models and correlation coefficients, are appropriate for the research questions addressed. Furthermore, the interpretation of the findings is also conducted appropriately.

2. Has the statistical analysis been performed appropriately and rigorously?

Yes. The study employs descriptive statistics to characterize the study population and utilizes various statistical tests, including Welch’s test, one-way ANOVA, chi-square test, and Kruskal–Wallis test, for comparative analysis. Generalized linear models (GLM) are used for regression analysis to explore the association between variables while adjusting for potential confounders, such as age, sex, and height. Spearman correlation coefficients are used to analyze the relationship between two variables, such as the correlation between kynurenines and glomerular filtration rate (GFR). Spearman correlation is suitable for assessing the strength and direction of monotonic relationships between variables. The significance level is set at p < 0.05, and statistical significance is reported for all tests.

3. Have the authors made all data underlying the findings in their manuscript fully available?

Yes. The authors have stated that all data are fully available without restriction, and the reviewer has confirmed this availability during the review process.

4. Is the manuscript presented in an intelligible fashion and written in standard English?

Yes. The manuscript is generally presented in an intelligible fashion and written in standard English. It exhibits clear writing, and follows a logical structure typical of scientific papers, which aids in reader navigation. Technical terminology common in biomedical research is defined or explained in context, enhancing understanding for readers. Effective use of tables and figures helps to present data clearly, with clear labels and legends aiding interpretation. The manuscript also demonstrates proficiency in standard English grammar and language usage, with correct sentence structures and minimal errors.

However, to improve the overall quality of the manuscript, several revisions are suggested.

a) Study Design

It is preferable to state the study design in the methods section rather than in the introduction. Additionally, since the design could also be considered as a cohort, providing further clarification regarding the decision not to categorize the study as a cohort study despite the inclusion of a control group would be beneficial. This clarification will aid in the accurate interpretation of the findings.

b) Setting and Participants

- It would be beneficial to place the information about the study settings before the study participants. Additionally, providing relevant background information about the study location would enhance the context of the research.

- To prevent any confusion regarding the recruitment period (line 84), specifying "18 September 2002 to 11 June 2009” would be preferable.

- Please consider adding specific data collection dates for each study phase to provide a comprehensive timeline.

- Furthermore, including details about sample size calculations and sampling techniques is essential for transparency and methodological rigor.

c) Statistical Methods

While the statistical analysis appears appropriate and rigorous, it would be beneficial for the authors to provide additional explanation regarding missing data handling, handling of participants lost to follow-up, and analytical methods considering the sampling strategy, if applicable. Alternatively, if these approaches were not employed, the authors should explain the rationale behind their decision.

d) Discussion

The discussion offers an insightful interpretation of the study findings and their alignment with existing literature. However, the authors could delve further into potential biases arising from the study design and strategies employed to minimize them. Additionally, discussing the external validity (generalizability) of the study findings would provide a more comprehensive understanding of its implications.

By making the necessary adjustments to address the raised points, I am confident that this manuscript could greatly contribute to the journal. I hope the authors will find my feedback helpful in enhancing the quality of their work.

Reviewer #2: PONE-D-24-08103

Kynurenines and aerobic exercise capacity in chronic kidney disease: a cross-sectional and longitudinal study

This is an interesting cross-sectional and longitudinal study, properly conducted, that analyzes the association between aerobic exercise capacity and kynurenines in non-dialysis patients with chronic kidney disease, aimed to show evidence that aerobic exercise capacity would be negatively associated with plasma levels of kynurenine, kynurenine/amino acid tryptophan and kynurenic acid in chronic kidney disease.

However, the relevance of these results in the clinic in patients is still underway. Furthermore, it has not been taken in account some other different or additional comorbidities to chronic kidney disease.

Nevertheless, this study may establish an interesting basis or support to future analysis in this sense, including studies on mitochondrial dysfunction, as suggested by the authors themselves, and/or potential molecular mechanisms involved in these processes.

Reviewer #3: It is an interesting study not previously published with these characteristics or variables. It would only improve to do another study by increasing the sample and controls in another area and different ethnic group.

6. PLOS authors have the option to publish the peer review history of their article (what does this mean?). If published, this will include your full peer review and any attached files.

Reviewer #1: No

Reviewer #2: No

Reviewer #3: No

---

## [Author Response · Author response to Decision Letter 0]

4 Sep 2024

Comment: General

• We note that you have included the phrase “data not shown” in your manuscript. Unfortunately, this does not meet our data sharing requirements. PLOS does not permit references to inaccessible data. Please add a citation to support this phrase or upload the data that corresponds with these findings to a stable repository (such as Figshare or Dryad) and provide and URLs, DOIs, or accession numbers that may be used to access these data. Or, if the data are not a core part of the research being presented in your study, we ask that you remove the phrase that refers to these data.

Response from authors: We agree with this, and we have removed the phrase “data not shown” and instead added the following sentence in the Results section: “TRP was not correlated with age in any of the groups (rs = -0.07 and p = 0.6 for CKD 2-3, rs = -0.3 and p = 0.07 for CKD 4-5, rs = -0.2 and p = 0.2 for the controls)”. (page 11, line 227-229)

Review Comments to the Author

Reviewer #1: 

Reviewer #1: Thank you for the opportunity to review this manuscript. Overall, this study provides a thorough examination of the relationship between aerobic exercise capacity and kynurenines in non-dialysis chronic kidney disease (CKD) patients. I appreciate the authors' dedication to undertaking this significant study.

1. Is the manuscript technically sound, and do the data support the conclusions?

Yes. Overall, the manuscript appears technically sound, and the data presented support the conclusions drawn by the authors. The study design provides robust evidence for the association between plasma kynurenine levels and aerobic exercise capacity in non-dialysis CKD patients. The statistical analyses performed, such as general linear models and correlation coefficients, are appropriate for the research questions addressed. Furthermore, the interpretation of the findings is also conducted appropriately.

2. Has the statistical analysis been performed appropriately and rigorously?

Yes. The study employs descriptive statistics to characterize the study population and utilizes various statistical tests, including Welch’s test, one-way ANOVA, chi-square test, and Kruskal–Wallis test, for comparative analysis. Generalized linear models (GLM) are used for regression analysis to explore the association between variables while adjusting for potential confounders, such as age, sex, and height. Spearman correlation coefficients are used to analyze the relationship between two variables, such as the correlation between kynurenines and glomerular filtration rate (GFR). Spearman correlation is suitable for assessing the strength and direction of monotonic relationships between variables. The significance level is set at p < 0.05, and statistical significance is reported for all tests.

3. Have the authors made all data underlying the findings in their manuscript fully available?

Yes. The authors have stated that all data are fully available without restriction, and the reviewer has confirmed this availability during the review process.

4. Is the manuscript presented in an intelligible fashion and written in standard English?

Yes. The manuscript is generally presented in an intelligible fashion and written in standard English. It exhibits clear writing, and follows a logical structure typical of scientific papers, which aids in reader navigation. Technical terminology common in biomedical research is defined or explained in context, enhancing understanding for readers. Effective use of tables and figures helps to present data clearly, with clear labels and legends aiding interpretation. The manuscript also demonstrates proficiency in standard English grammar and language usage, with correct sentence structures and minimal errors.

However, to improve the overall quality of the manuscript, several revisions are suggested.

Comment : a) Study Design

It is preferable to state the study design in the methods section rather than in the introduction. 

Additionally, since the design could also be considered as a cohort, providing further clarification regarding the decision not to categorize the study as a cohort study despite the inclusion of a control group would be beneficial. This clarification will aid in the accurate interpretation of the findings.

Response from authors: Indeed, we agree with these comments. We have now included the study design information in the Methods section and rephrased the last sentence in the introduction using “analyses” instead of “design”. 

Yes, our study should be considered as a cohort study and we added this information to the Methods section as following: 

Page 5, line 80-82: “The current study population is part of the PROGRESS 2002 study, a single-center, prospective observational cohort study (4, 20-22) conducted at Karolinska University Hospital in Stockholm, Sweden. Cross-sectional and longitudinal analyses were performed. “

Comment: b) Setting and Participants

- It would be beneficial to place the information about the study settings before the study participants. 

Additionally, providing relevant background information about the study location would enhance the context of the research.

Response from authors: We added information about the study location as “Karolinska University Hospital in Stockholm, Sweden” in the beginning of the Methods section, showing that the study was undertaken at a hospital in a clinical setting, in Stockholm, Sweden (page 5, line 82).

Comment: To prevent any confusion regarding the recruitment period (line 84), specifying "18 September 2002 to 11 June 2009” would be preferable.

Response from authors: Thank you, according to the suggestions we have added the dates (page 5, line 89).

Comment: Please consider adding specific data collection dates for each study phase to provide a comprehensive timeline.

Response from authors: We have added: “Baseline data was collected for the CKD 2-3 group during 2002-2003, for CKD 4-5 during 2002-2009, and for the control group during 2004-2007”. (page 5, line 92-93)”. We also added: ”Blood sampling and exercise tests were conducted at the most a few days apart.” (page 5, line 84-85).

Comment: Furthermore, including details about sample size calculations and sampling techniques is essential for transparency and methodological rigor.

Response from authors: The sample size was based on the PROGRESS study, where the aim was to find correlations between progress rate of kidney disease in CKD 2-3 and different markers, so it was not specifically calculated for this study. We have therefore added this sentence to the Limitations part: “No specific sample size calculation was performed, since the current study of the association between ExCap and kynurenines was of an exploratory character”. (page 16, line 406-408).

We have added more information regarding sampling techniques in the Methods section: “The control group was recruited either by random selection from the Swedish Total Population Register or by advertisement at the Karolinska University Hospital website.” (page 5, line 89-91). We also added information that the patients were consecutively recruited during outpatient visits: “The patients were consecutively recruited from the Department of Nephrology at Karolinska University Hospital from September 18th 2002–June 11th 2009 during outpatient visits.” (page 5, line 87-89).

Comment: c) Statistical Methods

While the statistical analysis appears appropriate and rigorous, it would be beneficial for the authors to provide additional explanation regarding 

missing data handling, handling of participants lost to follow-up, and analytical methods considering the sampling strategy, if applicable. Alternatively, if these approaches were not employed, the authors should explain the rationale behind their decision.

Response: Thank you for your comments regarding the statistics. We have described the sampling strategy for both the CKD patients and controls in more details, in response to an earlier question (see Methods, page 5, line 87-91)

Regarding missing data at baseline, this mostly concerns outliers in kynurenine, kynurenic acid and tryptophane analyses, that we excluded. We chose to exclude extreme outliers based on the following criteria, as described in the manuscript: “a data point that lies more than three times the interquartile range above the third quartile or below the first quartile” (page 8, line 162-164). We strongly believed that the outliers were caused by measurement error and not a biological variation, and therefore were excluded. In our Figure 1, number or outliers for each variable are clarified. 

Regarding missing data at follow-up/lost to follow-up for the 5-year data analysis, the general linear model does not control for this, as it examines the association between variables. However, we have analysed differences in the baseline variables exercise capacity (ExCap), GFR and age between individuals who did and who did not attend the exercise test at the 5-year follow-up. We did this separately for the CKD 2-3 and the control group. There were 9 (17%) individuals from the original CKD 2-3 group and 14 (26%) from the control group who did not attend the exercise test at follow-up. There were no significant differences between the individuals who did and did not attend follow-up in CKD 2-3, (p=0.09 for ExCap, 0.2 for age, 0.6 for GFR). For the control group, there was a borderline difference in ExCap (p=0.051),. but no significant differences in age (p=0.06) or GFR (p=0.1). We have added a supplementary file with these results (S5). We have also added the following sentence in the Results section: “Subjects lost to follow-up, in both the CKD 2-3 and control group, did not differ significantly regarding age, GFR and ExCap from those attending follow-up (see supplementary file 5)” (page 9, line 273-275)

We would also like to mention that in a previous publication we analysed the changes over 5 years in several variables in this cohort, including exercise capacity (reference 20 in the reference list). We then used a linear mixed model which considers missing data and lost-to-follow up, and exercise capacity did not change significantly in CKD 2-3 or in the control group from baseline to the 5-year follow-up. 

Comment - d) Discussion

The discussion offers an insightful interpretation of the study findings and their alignment with existing literature. However, the authors could delve further into potential biases arising from the study design and strategies employed to minimize them.

Response from authors: We agree that there are potential biases, which we have now added information about in the Limitations part:

“The controls were generally recruited at a later stage than the CKD group, however the examinations were carried out according to the same protocols during the whole study period. Recruitment bias may be a possibility in studies assessing exercise capacity, as e.g. sedentary patients are less likely to choose to take part.” (page 16, line 410-412).

Additionally, discussing the external validity (generalizability) of the study findings would provide a more comprehensive understanding of its implications.

Response from authors: Thank you for this comment, we have added to the manuscript page 16, line 395-399: “Our findings could be applicable to the broader population of non-dialysis CKD as it includes a large span of GFR, ranging from mild to severe disease, and the cohort also includes different etiologies of CKD. However, the follow-up data is limited to the CKD 2-3 group, and the results might be more applicable to this population”. 

By making the necessary adjustments to address the raised points, I am confident that this manuscript could greatly contribute to the journal. I hope the authors will find my feedback helpful in enhancing the quality of their work.

Reviewer #2: PONE-D-24-08103

Kynurenines and aerobic exercise capacity in chronic kidney disease: a cross-sectional and longitudinal study

This is an interesting cross-sectional and longitudinal study, properly conducted, that analyzes the association between aerobic exercise capacity and kynurenines in non-dialysis patients with chronic kidney disease, aimed to show evidence that aerobic exercise capacity would be negatively associated with plasma levels of kynurenine, kynurenine/amino acid tryptophan and kynurenic acid in chronic kidney disease.

However, the relevance of these results in the clinic in patients is still underway. Furthermore, it has not been taken in account some other different or additional comorbidities to chronic kidney disease.

Nevertheless, this study may establish an interesting basis or support to future analysis in this sense, including studies on mitochondrial dysfunction, as suggested by the authors themselves, and/or potential molecular mechanisms involved in these processes.

Response from authors: Thank you for your insightful comments. You are right that we still do not know the relevance of the results in the clinics. We have therefore added to clinical implications: “This study did not test the clinical relevance of the results and larger studies are needed to assess that”. (page 15, line 387-388).

Regarding comorbidities we have added: “Even though the cohort did not have severe cardiovascular disease, some of the patients had diabetes. While it would be interesting to explore the influence of comorbidities on the results, this would also require a larger study” (page 15, line 388-390).

We hope that you will find the revised version of our manuscript acceptable for publication. 

Kind Regards, 

Helena Wallin (first author)

---

## [Decision Letter · Decision Letter 1]

14 Nov 2024

PONE-D-24-08103R1Kynurenines and aerobic exercise capacity in chronic kidney disease: a cross-sectional and longitudinal studyPLOS ONE

Dear Dr. Wallin,

Thank you for submitting your manuscript to PLOS ONE. After careful consideration, we feel that it has merit but does not fully meet PLOS ONE’s publication criteria as it currently stands. Therefore, we invite you to submit a revised version of the manuscript that addresses the points raised during the review process.

We look forward to receiving your revised manuscript.

Kind regards,

Emma Campbell, Ph.D

Staff Editor

PLOS ONE

on behalf of 

Yousef Khazaei Monfared

Academic Editor

PLOS ONE

Additional Editor Comments:

Hi Dr. Wallin,

I am pleased to inform you that your paper, after being reviewed, has been deemed to have the merit for acceptance. Please address Reviewer 4's comments as you prepare the final version.

Best regards,

Reviewers' comments:

Reviewer's Responses to Questions

**Comments to the Author**

1. If the authors have adequately addressed your comments raised in a previous round of review and you feel that this manuscript is now acceptable for publication, you may indicate that here to bypass the “Comments to the Author” section, enter your conflict of interest statement in the “Confidential to Editor” section, and submit your "Accept" recommendation.

Reviewer #1: All comments have been addressed

Reviewer #4: (No Response)

2. Is the manuscript technically sound, and do the data support the conclusions?

Reviewer #1: Yes

Reviewer #4: Yes

3. Has the statistical analysis been performed appropriately and rigorously? 

Reviewer #1: Yes

Reviewer #4: Yes

4. Have the authors made all data underlying the findings in their manuscript fully available?

Reviewer #1: Yes

Reviewer #4: Yes

5. Is the manuscript presented in an intelligible fashion and written in standard English?

Reviewer #1: Yes

Reviewer #4: Yes

6. Review Comments to the Author

Reviewer #1: Thank you for thoroughly addressing the comments and revising the manuscript. The changes have clearly improved the article's quality, making it more informative and valuable to the field. I confirm that the manuscript meets the journal's standards for data availability, statistical integrity, and readability.

Reviewer #4: Dear Editor,

The original cross-sectional and longitudinal research paper entitled “Kynurenines and aerobic exercise capacity in chronic kidney disease: a cross-sectional and longitudinal study” is well-written, structured and developed by Wallin et al. in suitable English with a clear structure. They implemented a cross-sectional and longitudinal study to evaluate the association between aerobic exercise capacity and the plasma levels of accumulation of tryptophan and kynurenines in chronic kidney disease. They finally found that aerobic exercise capacity is inversely associated with the plasma levels of accumulation of tryptophan and kynurenines in chronic kidney disease. The findings are interesting, the methodology is suitable and novel, and the results have been appropriately discussed. I have two main concerns regarding this manuscript. First, there are several data in Table form such as Tables 3, 4 and 6 which can be considered and assigned as supplementary data. Second, the presentation strategy in this paper is not suitable. The paper is not illustrative. I highly suggest to summery the data in Tables into some appropriate illustration and present them in one or two figures. After addressing these major revisions, I may also re-review the paper.

7. PLOS authors have the option to publish the peer review history of their article (what does this mean?). If published, this will include your full peer review and any attached files.

Reviewer #1: No

Reviewer #4: **Yes: **Babak Pakbin

---

## [Author Response · Author response to Decision Letter 1]

20 Dec 2024

Thank you for your valuable comments regarding the presentation. We have made several changes to the presentation, which we believe have improved the manuscript:

1) We have replaced table 2 with six subfigures (Figure 2 a-f) which contain the same information as table 2 did but in a more illustrative way. Figure 2 caption can be found on page 11 and figure 2 is attached.

Figure 2 a-f. Box plots of peak workload, TRP, and kynurenines in CKD 2–3, CKD 4–5 and controls at baseline. Whiskers represent highest through lowest values, boxes the interquartile range and circles individual values. a: p-value by one-way ANOVA with Tukey post hoc test, b-f: p-values by one-way ANOVA with Welch’s test and Dunnett’s T3 post hoc test. KYN = kynurenine, KYNA = kynurenine acid, TRP = tryptophan. Number of subjects: 45-53.

2) We have added one additional figure - Figure 3 a and b. This figure replaces the old table 3 and 6, which have been modified and moved to supplementary information, see supplementary table S1 and S2. We think that Figure 3 summarizes the most important findings. Figure 3 caption can be found on page 12 and figure 3 is attached.

Figure 3 a) Association between kynurenines, TRP and aerobic exercise capacity (expressed as peak workload) in CKD 2–5. b) Association between the change in kynurenines, TRP and aerobic exercise capacity over 5 years in CKD 2-3. *p < 0.05, **p < 0.01, ***, p < 0.001 denotes significant inverse association. Assessed by GLM (generalised linear model) with age, sex and height included in all baseline models., KYN = kynurenine, KYNA = kynurenic acid, TRP = tryptophan, Wmax = peak workload on maximal exercise test.

3) We have also constructed one new table (table 2), which replaces the previous table 4 and 5. Some data have also been removed. We think the data in table 2 should be kept in the main manuscript and not in the supplementary information. Table 2 can be found on page 13. Below is the title of table 2.

Table 2. Aerobic exercise capacity, kynurenines and TRP at baseline and at 5 years follow-up, and a comparison between CKD 2–3 and controls regarding the change in these variables. 

4) We have changed the figure caption and design of figure 4 (previously figure 2). The caption can be found on page 12 and figure 4 is attached.

Figure 4. The relationship between aerobic exercise capacity expressed as peak workload as a percentage of predicted peak workload and KYN/TRP ratio, divided by CKD 2-3, CKD 4-5 and controls. KYN = kynurenine, TRP = tryptophan.

5) We have made several changes in the text in the results section due to the new figures and tables. They are visible in the tracked changes version of the manuscript. The most important changes are that we added the values of GFR and hemoglobin at baseline and at follow-up for both CKD and controls (page 294, line 286-291 in tracked changes manuscript). This information could be found in a table in the previous version. 

To summarize, our main data are now presented as:

Figure 1 – Study population

Figure 2 - Box plots of peak workload, TRP, and kynurenines in CKD 2–3, CKD 4–5 and controls at baseline

Figure 3 a) and b). 3 a) Association between TRP and kynurenines and baseline exercise capacity in CKD 2–5. 3 b) The association between the change in those variables over 5 years.

Figure 4 – A graph of the relationship between aerobic exercise capacity expressed as achieved workload as a percentage of predicted peak workload and KYN/TRP ratio, divided by CKD 2-3, CKD 4-5 and controls.

Table 1 - Subject characteristics for the whole study population

Table 2 - Aerobic exercise capacity, kynurenines and TRP at baseline and at 5 years follow-up, and a comparison between CKD 2–3 and controls regarding the change in these variables.

Supporting information: 

S1 table: Association between aerobic exercise capacity and TRP, kynurenines, TRP and GFR in CKD 2–5 at baseline (details from the generalised linear models).

S2 table: Predictors of the change in aerobic exercise capacity from baseline to 5-year follow-up in CKD 2–3 ((details from the generalised linear models).

General comment from authors to reviewers and editor:

We have removed the supplementary file that contained analyses on subjects lost to follow-up and instead added in the manuscript that p-values were > 0.05 (page 14, line 294-296 in tracked changes manuscript) for differences between subjects attending follow-up and subjects lost to follow-up. We have also removed a supplementary file that contained detailed output from GLMs, as we do not think the information need to be provided there. They are available upon request. 

We hope that you find that the adjustments have improved the manuscript and thank you for reviewing it. 

Kind Regards,

Helena Wallin, main author

---

## [Editor Report · Decision Letter 2]

23 Dec 2024

Kynurenines and aerobic exercise capacity in chronic kidney disease: a cross-sectional and longitudinal study

PONE-D-24-08103R2

Dear Dr. Helena Wallin,

We’re pleased to inform you that your manuscript has been judged scientifically suitable for publication and will be formally accepted for publication once it meets all outstanding technical requirements.

Kind regards,

Yousef Khazaei Monfared

Academic Editor

PLOS ONE
---

## [Editor Report · Acceptance letter]

2 Jan 2025

PONE-D-24-08103R2 

PLOS ONE

Dear Dr. Wallin, 

I'm pleased to inform you that your manuscript has been deemed suitable for publication in PLOS ONE. Congratulations! Your manuscript is now being handed over to our production team.

Kind regards, 

on behalf of

Dr. Yousef Khazaei Monfared 

Academic Editor

PLOS ONE